# Using Hawkes Processes to model imported and local malaria cases in near-elimination settings

H. Juliette T. Unwin[1]*, Isobel Routledge[1,2], Seth Flaxman[3], Marian-Andrei Rizoiu[4], Shengjie Lai[5], Justin Cohen[6], Daniel J. Weiss[7,8,9], Swapnil Mishra[1], Samir Bhatt[1]

**1** MRC Centre for Global Infectious Disease Analysis, Jameel Institute for Disease and Emergency Analytics, Imperial College, London, United Kingdom, **2** Department of Medicine, University of California, San Francisco, California, United States of America, **3** Department of Mathematics, Imperial College, London, United Kingdom, **4** Data Science Institute, University of Technology Sydney, Sydney, Australia, **5** WorldPop, School of Geography and Environmental Science, University of Southampton, Southampton, United Kingdom, **6** Clinton Health Access Initiative, Boston, Massachusetts, United States of America, **7** Malaria Atlas Project, Big Data Institute, Nuffield Department of Medicine, University of Oxford, Oxford, United Kingdom, **8** Telethon Kids Institute, Perth Children's Hospital, Nedlands, Western Australia, Australia, **9** Curtin University, Bentley, Western Australia, Australia

* h.unwin@imperial.ac.uk

**Data Availability Statement:** Fitting and simulation code is available on GitHub: https://github.com/mrc-ide/epihawkes and model outputs to recreate the figures from Harvard Dataverse:

## Abstract

Developing new methods for modelling infectious diseases outbreaks is important for monitoring transmission and developing policy. In this paper we propose using semi-mechanistic Hawkes Processes for modelling malaria transmission in near-elimination settings. Hawkes Processes are well founded mathematical methods that enable us to combine the benefits of both statistical and mechanistic models to recreate and forecast disease transmission beyond just malaria outbreak scenarios. These methods have been successfully used in numerous applications such as social media and earthquake modelling, but are not yet widespread in epidemiology. By using domain-specific knowledge, we can both recreate transmission curves for malaria in China and Eswatini and disentangle the proportion of cases which are imported from those that are community based.

## Author summary

This paper introduces a mathematically well-founded method for infectious disease outbreaks known as Hawkes Processes. These semi-mechanistic models are relatively new to the infectious diseases toolkit and enable us to combine disease specific information such as the infectious profile with statistical rigour to recreate temporal disease transmission. We show that these methods are very suited to modelling malaria in communities close to eliminating malaria—in particular China and Eswatini—where we are able to disentangle the contribution of exogenous (external) transmission and endogenous (person-to-person) transmission. This is particularly important for developing policies when counties are approaching elimination.

https://doi.org/10.7910/DVN/YPRLIL. The anonymised China data set comes from Routledge et al. (2020) Plos Comp Bio and can be found in the dataverse repository. The Eswatini data set comes from Reiner Jr et al. (2015) elife and requests should be directed to Robert C Reiner (rcreiner@indiana.edu), the corresponding author of the elife paper, or the Eswatini Ministry of Health (http://www.gov.sz/index.php/ministries-departments/ministry-of-health).

**Funding:** HJTU is funded by Imperial College London through an Imperial College Research Fellowship grant. SB acknowledges funding from the NIHR BRC Imperial College NHS Trust Infection themes (RDA02), the Academy of Medical Sciences Springboard award (SBF004/1080) and the Bill and Melinda Gates Foundation (CRR00280). HJTU, SM, IR and SB acknowledge joint centre funding (reference MR/R015600/1) by the UK Medical Research Council (MRC) and the UK Department for International Development (DFID) under the MRC/DFID Concordat agreement and is also part of the EDCTP2 programme supported by the European Union. MAR acknowledges funding from Facebook Research under the Content Policy Research Initiative grants, and the Defence Science and Technology Group of the Australian Department of Defence. The funders had no role in study design, data collection and analysis, decision to publish, or preparation of the manuscript.

**Competing interests:** The authors have declared that no competing interests exist.

## Introduction

Modelling infectious disease transmission is an important tool for monitoring outbreaks and developing public policy to limit the spread of the disease. One common source of data available during these types of outbreaks are line lists, or case counts, from surveillance systems. These define the time at which patients are infected, along with other epidemiological information such as the sex, age and symptoms of the patient, locations they were infected or live and if they have travelled recently. An ideal model would combine all the information available from the line lists with disease-specific mechanisms developed by experts of the disease to recreate case counts over time and accurately predict future behaviour. Traditionally, SIR (Susceptible—Infected—Recovered) type models, such as the seminal Kermack-McKendrick model [1], or individual-based models (for example [2] and [3]) have been used to model disease outbreaks. These methods encode well-known disease-specific mechanisms and can produce very good fits to data. However, they can require large amounts of data to produce these accurate fits, are cumbersome and computationally demanding to simulate from and difficult to forecast with. Therefore, there is scope to develop new methods and software to simulate outbreak behaviour. An alternative method proposed by Routledge et al. [4, 5] estimates temporal and spatial reproduction numbers by studying information diffusion processes in the form of network models, which reconstruct information transmission using known or inferred times of infection in a Bayesian framework [6]. These methods provide an adaptable framework to integrate multiple data types at different scales and identify missing data or external infection sources, but require very good data sets to accurately be able to predict from the models [6, 7].

SIR models can be linked to a well known statistical point process called Hawkes Processes [8], which we propose is a better alternative to model infectious disease outbreaks if the data is of high enough fidelity. These processes are semi-mechanistic, so give us the ability to encode disease specific information such as serial interval and incubation period, but are easier and computationally cheaper to simulate from and fit to data. Hawkes Processes model the intensity of infectious diseases by separating out contributions from exogenous and endogenous processes. The relative contributions of these two terms is disease specific and may have different levels of importance depending on the disease. The majority of transmission of Ebola is direct contact by human, and Kelly et al. [9] has recreated the Democratic Republic of Congo epicurve, or cases counts over time, using a Hawkes Process model with an endogenous term and a simple background transmission rate. However, there is a real need to correctly parameterise more complex exogenous terms for diseases such as malaria in near-elimination settings and cholera to reproduce and predict the spread of the disease accurately.

In this paper, we focus on applying Hawkes Processes to malaria in near-elimination settings, where current models may not be especially well suited [4, 10]. In 2016, the World Health Organisation identified 21 countries with the potential to eliminate malaria by 2020; seven of these countries (Algeria, China, El Salvador, Iran, Malaysia, Paraguay, and Timor-Leste) have eliminated malaria since that list was published [11]. Since then, The Lancet Commission has published research by Feachem et al. [12] suggesting that malaria eradication within a generation is ambitious, achievable and necessary, but there needs to be an immediate, firm, global commitment to achieving such eradication by 2050. This involves developing new methods for modelling near-elimination settings, which can accurately capture the behaviour and help governments and public health organisations implement the best interventions to bring their countries closer to elimination.

Malaria is a complex disease to model, especially in low transmission settings, where the entomological inoculation rate (number of infected bites a person receives) varies greatly due

to focal transmission and is potentially unstable due to sensitivity to heterogeneity in vector populations [4, 13, 14]. There are also inaccuracies in parasite prevalence rate estimations below 1-5% because a large sample size is necessary to accurately predict the proportion of the population with malaria [15]. We hypothesise that Hawkes Process models will help provide new insight into malaria transmission in these settings.

We introduce the traditional Hawkes Process in this paper and define the basic fitting and simulation algorithms, which use incidence data opposed to prevalence data. We then use our knowledge of malaria in near-elimination settings to tailor our exogenous and endogenous terms to best fit our data sets. We first evaluate our method for a simulated example and then for two case studies (China and Eswatini). These data sets include the time of symptom onset and if the case was reported as an importation through travel history. We apply our methods to recreate the case counts over time in our two data sets, show goodness of fit measures and forecast forward 35 days to evaluate our model predictions.

## Background

A uni-variate Hawkes Process is a self-exciting point process with a conditional intensity, $\lambda(t)$, defined as:

$$\lambda(t) = \mu(t) + \sum_{t>t_i}\phi(t - t_i),$$  (1)

where $\mu(t)$ is the exogenous time dependent contribution to the intensity from external disease importations and $\sum_{t>t_i}\phi(t - t_i)$ is the self exciting endogenous contribution representing person-to-person interactions [16]. Eq 1 means that the arrival of an event increases the likelihood of receiving a further event in the near future but that the importations are independent of all other events. Alternatively, a person getting infected increases the short term chance of other infections within the community, but people can also be infected independently from outside sources, such as zoonotic spillover or by travelling into the community already infected. The function $\phi(\cdot)$ is often referred to as the triggering kernel in the Hawkes Process literature and describes a parameter similar to the serial interval distribution, or the expected time between infection and subsequent transmission. The parameter $t_i$ refers to the times of the past events or in epidemiological applications, previous infections.

Similar to the simplest class of point processes, the Poisson Process [17], each event can be independently sampled from an intensity distribution. Unlike Poisson Processes, the intensity distribution of Hawkes processes is dependent on previous events because they are self-exciting, i.e. the occurrence of past events increases the likelihood of future events. The intensity of the Hawkes Processes is a stochastic function because it depends on event times which are random variables, however the Hawkes Process can be treated as a non-homogeneous Poisson Process between events. The methods have been used successfully to model numerous applications such as earthquakes [18], crime [19], financial time series [20] and social media [21–24]. However, although a few people now use Hawkes Processes for epidemiological modelling [9, 25–27], they are not common place methods in this field yet.

The link between Susceptible—Infected—Recovered (SIR) and Hawkes Process models has been shown by Rizoiu et al. [8] for finite population sizes. They generalise the Hawkes Process to HawkesN and show that these types of models are conceptually similar to SIR models. The time varying intensity function of HawkesN, $\lambda^H(t)$, is defined as

$$\lambda^H(t) = \left(1 - \frac{N_t}{N}\right)\left(\mu(t) + \sum_{t>t_i}\phi(t - t_i)\right)$$  (2)

where $N$ is the total population, $N_t$ is the number of infections that occurred before or at time $t$ (assuming immunity from the disease arises post infection) and, as before, $\mu(t)$ is the exogenous time dependent contribution to the intensity from external disease importations and $\sum_{t>t_i}\phi(t-t_i)$ is the self exciting endogenous contribution representing person-to-person interactions. This is similar to the Hawkes Process intensity in Eq (1) but also includes a population weighting term. Past events generate new events at a rate of $\phi(t)$ in HawkesN, which is analogous to the population adjusted infection rate $\frac{\beta S_t}{N}$ in the SIR models [1, 28], where $\beta$ is the infection rate, $S_t$ is the number of susceptible individuals at time $t$ and $N$ is the population size. Rizoiu et al. provide evidence that if the events in a HawkesN Process with parameters $\{\mu$ (background intensity), $\alpha$ (magnitude of infection kernel), $\delta$ (parameter controlling duration of infection), $N$ (size of population)$\}$ have the intensity $\lambda^H(t)$ and the new infections of a stochastic SIR model with parameters $\{\beta$ (infection rate), $\gamma$ (recovery rate), $N$ (population size)$\}$ follow a point process of intensity $\lambda^I(t)$, the expectation of $\lambda^I(t)$ over all event times $\mathcal{T} = \tau_1, \tau_2, \ldots$ is equal $\lambda^H(t)$:

$$\mathbb{E}_{\mathcal{T}}[\lambda^I(t)] = \lambda^H(t), \tag{3}$$

when $\mu = 0$, $\beta = \alpha$ and $\gamma = \theta$. In this paper we consider the univariate Hawkes Process (as described by Eq (1)), instead of HawkesN, because we consider near-elimination malaria outbreaks where we assume an infinite susceptible population. This means that $N_t/N$ is small.

## Methods

Hawkes Processes are semi-mechanistic because we can incorporate disease specific information into our infection mechanism. Instead of using the traditional exponential kernel as explained in S1 Text, we propose using a Rayleigh kernel of the form

$$\phi(t-t_i) = \alpha * (t-t_i)e^{-\delta*(t-t_i)^2/2} \quad \forall t > t_i \tag{4}$$

to model the within country transmission of malaria, where $\alpha \geq 0$ controls the magnitude of the force of infection from an infected individual and $\delta \geq 0$ controls the length of the infectious period. We choose this kernel because a person is not most infectious immediately after they are bitten by a mosquito. This kernel is little used in applications of Hawkes Process but has been suggested by Wallinga et al. [29] Gomez et al. [30] and Ding et al. [31] and has already been used to represent the serial interval in malaria models [4]. We also used malaria domain specific knowledge to impose a delay between the mosquito biting an infectious person and become infectious and the person that mosquito going on to bite becoming infectious. Therefore, our kernel is

$$\phi(t-(t_i+\Delta)) = \alpha * (t-(t_i+\Delta))e^{-\delta*(t-(t_i+\Delta))^2/2} \quad \forall t > t_i + \Delta, \tag{5}$$

where $\Delta > 0$ represents the delay. This delay is novel and requires modifications to be made to the usual simulation approach; this is explained further below. We fit $\alpha$ and $\delta$ in our model and assume the value of $\Delta = 15$ days from literature [5]. The incorporation of a delay is still necessary despite our infection times being the time of symptoms onset due to the role of the mosquito. There is still a delay before the second person can onset due to the time it takes for the mosquito to pass on the infection.

We also propose using a more complex time varying exogenous term than is found in literature (e.g. [23] and [32]) to capture the behaviour of the imported malaria cases. Our $\mu$ has the

form

$$\mu(t) = \max\left(A + Bt + M\cos\left(\frac{2\pi t}{p}\right) + N\sin\left(\frac{2\pi t}{p}\right), 0\right), \tag{6}$$

where $p$ = 365.25 and $A$, $B$, $M$ and $N$ are constants that are fitted from data. This captures the linear decrease in exogenous events that we would expect in a malaria elimination setting along with the yearly fluctuating seasonality trends that often are associated with malaria. The $M$ and $N$ parameters will contribute less to the importations in areas with little or no seasonality. Unfortunately this also leads to a more complicated simulation process because the sinusoidal terms cause $\mu$ to increase periodically and also can result in non-convexity in our log-likelihood [23], see below.

## Fitting Hawkes Processes

We use *optimx* from the optimx package [33] to minimise our log-likelihood and choose our optimal values for $\alpha$, $\delta$, $A$, $B$, $M$ and $N$. We provide the analytic directional derivatives of our log-likelihood in S2 Text, which we use as additional parameters to improve the efficiency of the *optimx* package. We calculate 95% confidence intervals for our parameters using the bootstrapping approach in Reinhart [34] and Sarma et al. [35]. We simulate 10,000 simulations following the procedure below and re-fit each set of parameters, ensuring that $T_{max}$ in our simulations is equal to or less than the last infection in our data set. The 95% confidence intervals are the 2.5% and 97.5% quantiles of the 10, 000 refits. We ensure our optimal parameter sets from re-fitting each simulation form a true minima and not a saddle point by refitting until all the eigenvalues from our hessian, evaluated at the optimal solution, are positive.

We use goodness of fit tests to evaluate our fits. First we consider how $\Lambda(t_i)$ varies with index of the event, $i$. Similar to Brown et al. [36], we define

$$\Lambda(t_i) = \int_0^{t_i} \lambda(t)\,\mathrm{d}t. \tag{7}$$

If the model fits well, the integral of the intensity evaluated at each event plotted against the index should lie along a straight line. We also use the time–rescaling theorem. According to this theorem, the difference in $\Lambda(t_i)$ between two subsequent events are independent exponential random variables with mean 1. We present Kolmogorov–Smirnov (KS) tests and quantile–quantile (Q–Q) plots as goodness of fit tests to assess the quality of our fits; the points should lie on a 45-degree line if the model is a good fit.

## Simulating from a complex intensity function

It is not trivial to simulate from our intensity function for two reasons. First, our kernel is not monotonically decreasing and, second, we impose a fluctuating exogenous term. Alternative cluster based methods for simulation e.g. Reinhart [34] could provide similar results to the algorithm we present below, but were not implemented here to allow further developments to be added to the kernel in due course and to reduce the complexity in the termination conditions.

The time of the maximum intensity from a single Rayleigh kernel at time $t$ is

$$t_{\text{max intensity}} = t + \frac{1}{\sqrt{\delta}}. \tag{8}$$

However, we can only place bounds on the time at which the intensity is maximum when it is comprised of multiple Rayleigh kernels, includes delays, $\Delta$, and has a time varying $\mu$; we did

not find an analytic solution. When $\mu = 0$ or is constant, the maximum lies between $t_{\text{last event}}$ and $t_{\text{last event}} + \frac{1}{\sqrt{\delta}} + \tau$, see S3 Text. These bounds have to be widened when considering non-monotonically decreasing exogenous terms because the maximum value of $\lambda$ can occur after $t_{\text{last event}} + \frac{1}{\sqrt{\delta}} + \tau$ if $\mu$ periodically increases. In Fig 1A and 1B the maximum of the kernel still

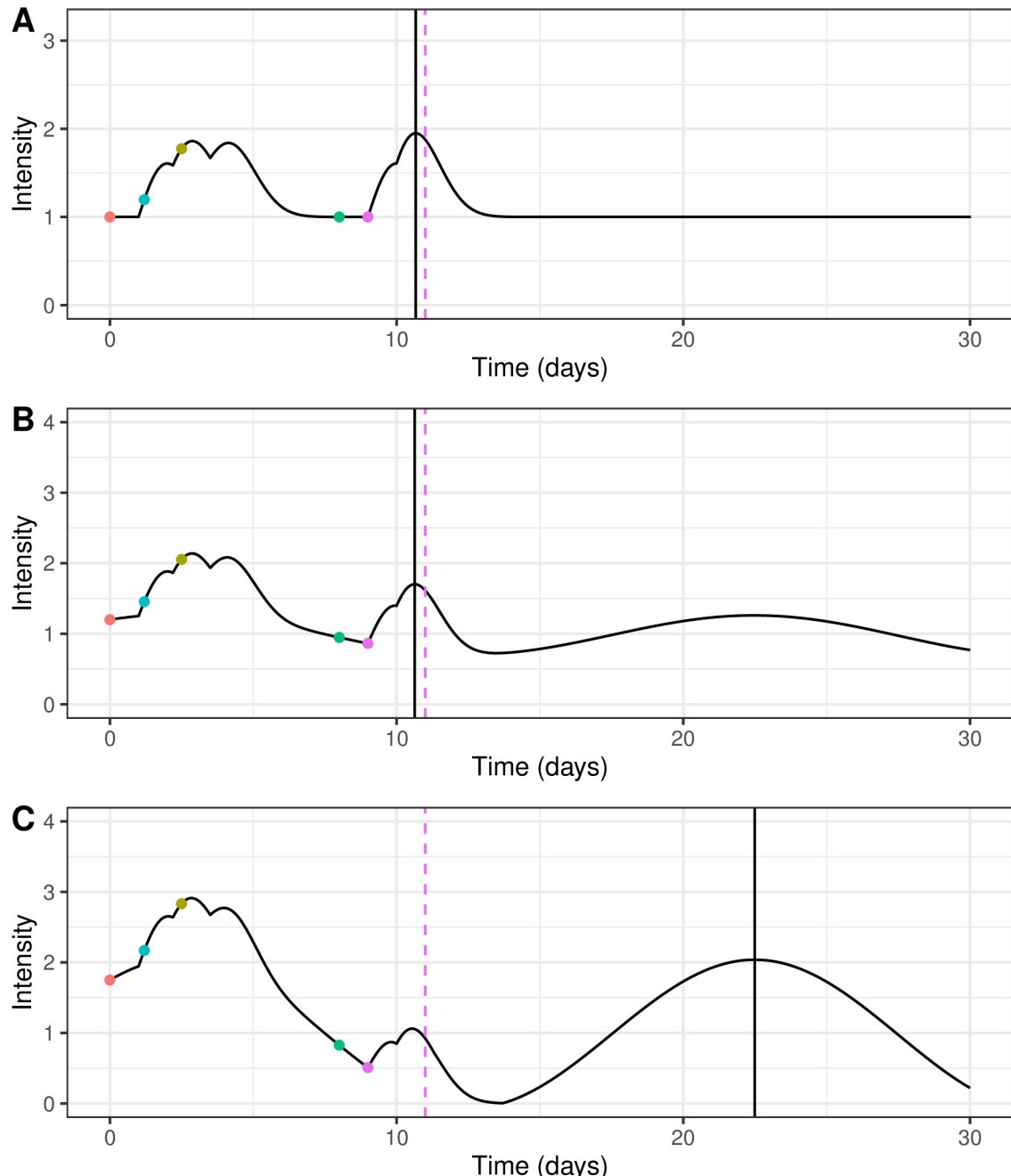

**Fig 1. Illustrative plot of intensity function for events occurring at times 0, 1.2, 2.5, 8 and 9 with kernel parameters $\alpha$ = 1.0 and $\delta$ = 1.0, a 1 day delay and a time varying $\mu$.** The coloured dots refer to different events or infections and the dashed pink line indicate the time of the theoretical maximum value of a single Rayleigh kernel at the last event time. The solid black line indicates the time of the maximum value of the kernel after the last event. Fig 1A shows a constant $\mu$ and Fig 1B and 1C show sinusoidal $\mu$ with a linear decrease of different magnitudes. The parameters for Eq (6) in each case are as follows: A—$A$ = 1; B—$A$ = 1, $B$ = −0.001, $M$ = 0.2, $N$ = 0.2 and $p$ = 20; C—$A$ = 1, $B$ = −0.001, $M$ = 0.75, $N$ = 0.75 and $p$ = 20. These parameters are only illustrative and do not reflect parameters we would expect real in malaria models.

lies between the last event and the time of the maximum value of the kernel at that time. However, Fig 1C shows that the maximum value of λ can occur outside that region and up until the maximum of the $\mu$ term. This is particularly important if the exogenous term dominates, which we predict happens in a near-elimination malaria settings.

We propose a new algorithm for finding the maximum of $\lambda(t)$. First we bound the times at which the maximum can occur; we calculate $\frac{1}{\sqrt{\delta}} + \tau$ and the time of the maximum value of the exogenous term, $t_{\mu \text{ max}}$ between the previous event and the final time of the simulation:

$$t_{\text{last event}} < t_{\max} < \max\left(t_{\text{last event}} + \frac{1}{\sqrt{\delta}}, t_{\mu \text{ max}}\right). \tag{9}$$

Since the intensity is the juxtaposition of multiple functions with known maximums, we can be sure that the maximum does not lie outside this bound. We then use a root finding algorithm similar to *uniroot.all* from the *rootSolve* package [37, 38] to locate all the roots of the derivative of the intensity. We do not know prior to the calculation how many roots there are so split the bound into a pre-defined number of sections and search for a sign change inside the interval. Once we have the times of these turning points, we evaluate them and find the maximum value of the intensity. This is summarised in Algorithm 1.

**Algorithm 1**: Algorithm for finding $\lambda^\star$

```
Bound the region in time which the maximum value of the intensity
occurs;
(a) The minimum value of the region is the time of the previous event
by definition t_min bound = t_last event;
(b) The maximum value of the region is the larger of the maximum time
of a single kernel at the last event time or the maximum value of μ
```
after the event $t_{\text{max bound}} = \max\left(t_{\text{last event}} + \frac{1}{\sqrt{\delta}}, t_{\mu \text{ max}}\right)$;
```
Compute the derivative of the intensity;
Find all roots of the derivative of the intensity or the turning
points of the intensity between t_min bound and t_max bound;
Evaluate the intensity at the turning points;
Select λ*;
```

## Simulated data

In this paper we first evaluate our model using simulated data. We simulate 10, 000 sets of events using Algorithms 1 and Supplementary Algorithm 1 for $\alpha = 0.017$, $\delta = 0.057$, $A = 0.400$, $B = 0.0001$, $M = 0.305$ and $N = -0.123$ with the 15 day delay. These were chosen because they are the optimum parameters that were fit to the Eswatini data set. We then use optimx to minimise our log-likelihood and find the optimal values of each of our simulations. We compare these fitted parameters to the initial parameters used for the simulation and evaluate our goodness of fit using the integral of our intensity evaluated at each event time, $\Lambda(t_i)$, and a KS plot.

We then consider the impact of under-reporting on the Hawkes Processes fits of our simulated data, which is common phenomenon in malaria case reporting. We choose to investigate this for our simulated data since we know these case series are complete, instead of inevitably missing cases in our two case study data sets especially in Eswatini. We implement this by randomly sampling different proportions (10% to 95%) of the first 1, 000 simulations computed above and compare the optimal fits from one initial set of parameters for each simulation to the original parameter sets. We can also estimate how the case reproduction number, $R_c$, varies with under-reporting by considering the branching factor of the Hawkes Process. The $R_c$ is equal to the reproduction number in the presence of a range of interventions and is defined in Hawkes Process literature as the average number of children events that result from one parent

event. This is derived in S4 Text for a Rayleigh kernel and is equal to the integral of the kernel between 0 and infinity:

$$R_c = \frac{\alpha}{\delta}. \qquad (10)$$

## Malaria case studies

In addition to simulated data, we fit our model to line lists of individuals with malaria in two countries over 1,000 days. We consider malaria cases caused by the *Plasmodium vivax* parasite between 1[st] January 2011 to 24[th] September 2013 in Yunnan Province, China [5] and all malaria cases between 24[th] February 2010 to 16[th] November 2012 in Eswatini [39]. These line lists only include people who attended a health clinic and received treatment. There are 2153 cases in our China and 627 cases in our Eswatini datasets. We assume all patients were treated as they were reported on our line list, which reduces the length of time they were infectious compared to an untreated malaria case. We chose these two data sets because the imported cases are labelled, although we do not use information about if a case was imported or local in our fitting process. Our cases are disaggregated by day, so we add right handed uniform jitter (ensuring the dates of each infection remain the same) to our times to ensure we have unique times for our events. This is a limitation of this method, but necessary for the Hawkes algorithm. We initialise the optimisation routine for fitting each data set from 10 different start points and select our final parameters to be the ones with the minimum negative log-likelihood.

We simulated 10,000 realisations of our Hawkes Process up to $T_{max} = 1,000$ using Algorithms 1 and Supplementary Algorithm 1, and our fitted parameters. From this we could recreate the daily number of cases and the epicurve, or cumulative cases, over time. We also simulated 10,000 realisations of just the $\mu$ term, or the endogenous cases only, which represented the imported malaria cases. We used the same algorithms as before, but set $\alpha = \delta = 0$ because we were not considering the cascade of infections from these importations at this time. We compared these simulations to a simple Hawkes Process model fitted using the traditional exponential kernel with a 15 day delay and a parametric growth model using the *growthrates* R package [40].

It is also possible to use Hawkes Process models for prediction. We can see how well our model fits future data by not fitting our model to all the available data. Instead we hold back the last portion of the epicurve and forecasting over the period of the withheld data. We simulated for 35 days more than we fit to so that we could investigate the predictive power of the model. Again, we compare our forecasts with those from the parametric growth rate model. All our Hawkes Process code is provided in the *epihawkes* package and available open source on GitHub[1].

## Results

### Simulated data

We show in Fig 2A and 2B (and Fig 3 (100% bar)) that we can recover the initial parameters from our 10,000 refits to our simulations. We find that a small number of our fits (under 2%) lie in a different parameter regime, which corresponds to a different minima in our non-convex log-likelihoood. This is a problem with having a non-convex optimisation surface, so care should be taken to ensure the parameter space is widely explored to maximise the chance of selecting the global minima. S1 Fig shows the un-magnified version of Fig 2B.

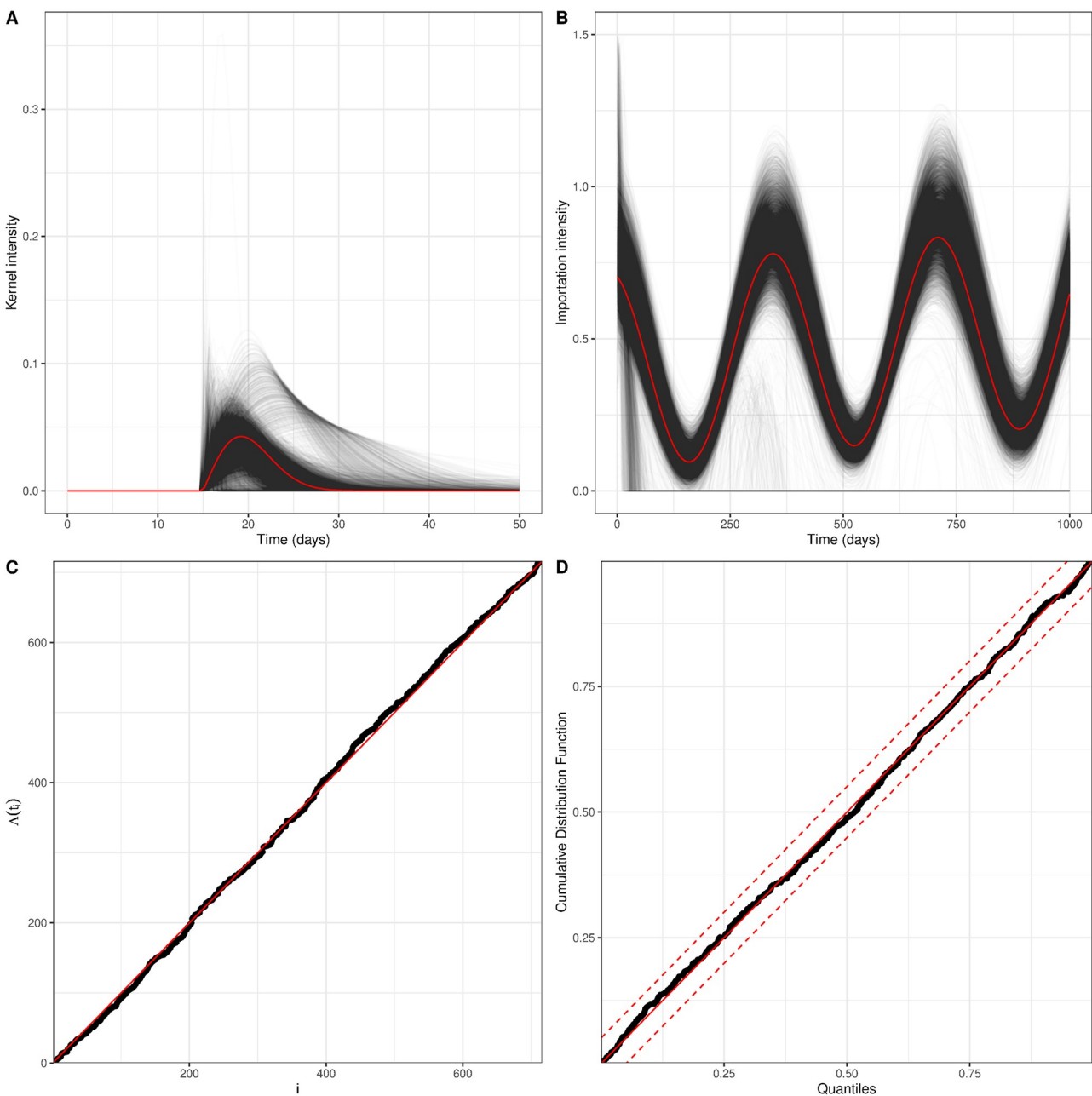

**Fig 2. Model fits for simulated data using parameters:** $\alpha = 0.017$, $\delta = 0.057$, $A = 0.400$, $B = 0.0001$, $M = 0.305$, $N = -0.123$ **and our fixed delay** $\Delta =$ **15.** Fig 2A shows the kernel from the true parameter in red with the kernels generated from the refits to each simulation in black. Fig 2B shows the how the exogenous term or importation intensity varies through time. The red line shows the importation intensity calculated from the initial parameters and the black lines shows the importation intensity calculated from the parameters fit from each simulation. This figure is magnified to show the region around the true value, but the un-magnified version is given in S1 Fig. Fig 2C shows the integral of the intensity evaluated at each event time plotted against the event index, for one simulation. The red solid line is $y = x$. Fig 2D shows the KS goodness of fit test from one simulation. The red solid line is $y = x$ and the red dashed lines represent the 95% confidence intervals.

Good performance of our fitting and simulation algorithms are suggested by our goodness of fit tests. The integral of our intensity, $\Lambda(t_i)$, evaluated at our event times plotted against the event index (Fig 2C) lie along a straight line, which suggests goodness of fit. In addition, we find that the black dots of a KS plot from a sample simulation in Fig 2D are approximately

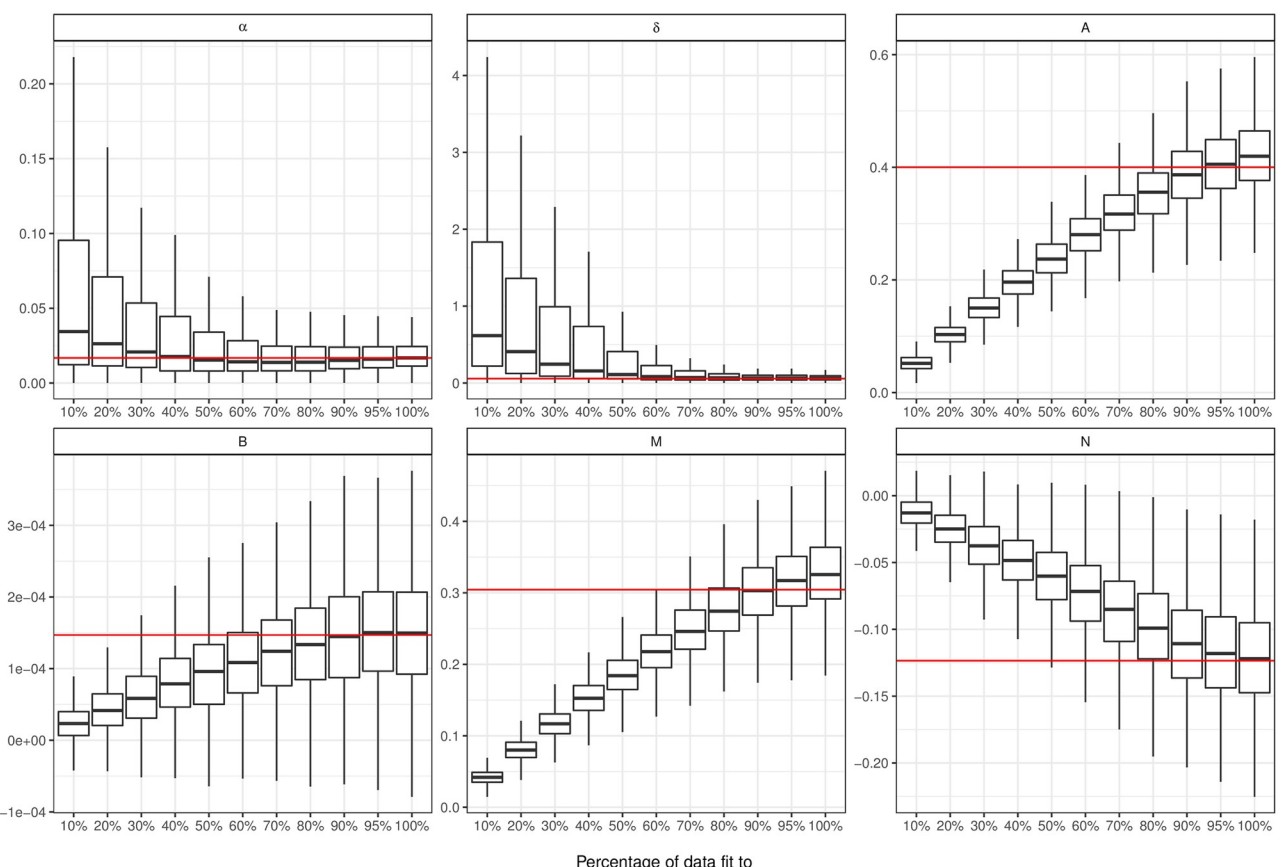

**Fig 3. Box and whisker plots showing the distribution of our fits to different proportions of the data.** Each of the parameters in our model is shown as a different plot. The red line is the true parameter used to generate our simulations and the box shows the interquartile range with the whiskers showing 1.5 times the interquartile range above and below the 25th and 75th percentile.

linear and all lie within the confidence intervals of the plot. This suggests that the difference in $\Lambda(t_i)$ between our simulated events are independent exponential random variables with mean 1, as expected.

We can also see that our Hawkes Process model is robust to some level of missing data, or under reporting. In Fig 3 we show that the true parameters lie within the interquartile range of all parameters for 90% of the data included in each fit, or 10% under reporting. We find that our kernel parameters are especially robust in most of the scenarios considered. This make sense because the kernel defines the biological process, with the background intensity changing to accommodate the missing cases. We find that these changes in parameters results in the median value of $R_c$ decreasing from 0.261 to 0.101 between 100% and 40% of cases reported being reported with overlapping confidence intervals, see S2 Fig. Our uncertainty is wide because our optimisation surface is non-convex and sometimes we arrive in a different local minima.

## Case studies

We can recreate our kernel and exogenous term using the optimal parameters returned by our fitting procedure. Fig 4A shows the fitted intensity for both China and Eswatini. The duration over which a person remains infectious, or where the intensity is greater than zero, is around 12 to 15 days for both China and Eswatini, but the individual contribution to the intensity

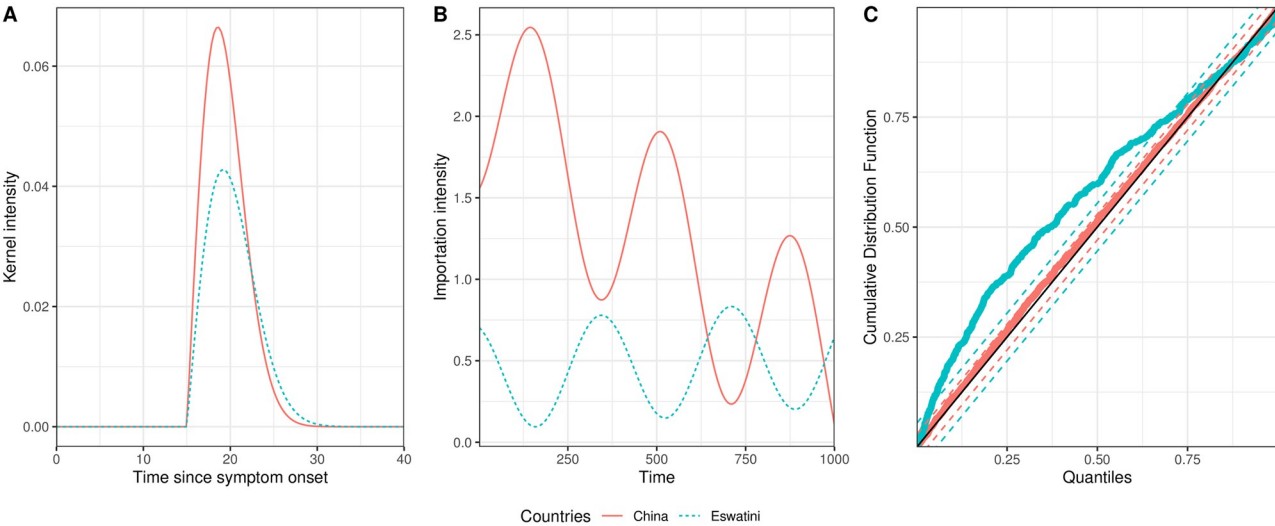

**Fig 4. Fitted endogenous and exogenous terms for the China and Eswatini data.** Fig 4A shows the fitted kernel intensity for a single infection, which corresponds Eq (5). Fig 4B shows how the exogenous terms vary through time. Fig 4C shows results from the Kolmogorov–Smirnov goodness of fit tests. The solid red lines and dots correspond to the China data and the dashed blue lines and dots correspond to the Eswatini data. The black solid line in Fig 4C is the line $y = x$ and the red and blue dashed lines are the 95% confidence intervals for the China and Eswatini data set respectively.

from one person is greater in China than Eswatini. The kernel, $\phi(t - t_i)$, is zero for the first 15 days, which corresponds to the delay in a secondary person becoming infectious due to the mosquito stage, even though we assume the infector is infectious at symptoms onset. Fig 4B shows how $\mu$ varies over time for our proposed model. This variation is very different between China and Eswatini; $\mu$ decreases significantly over the 1, 000 days in China, but the initial intensity is much lower in Eswatini and increases slightly. Using these parameters, we calculate the $R_c$ for China to be 0.39 [0.23 − 0.99] and Eswatini to be 0.30 [0.05 − 1.02], where the square brackets denote the 95% confidence intervals calculated through a boot strapping method. This cannot be calculated from the growth model, which we compare our subsequent results to. Uncertainty in all our model fit parameters are given in S1 and S2 Tables.

We see from our KS goodness of fit test (Fig 4C) that our fit to the China data is very good and lies within the red dashed confidence interval but our Eswatini fit is less good as we explain later. This pattern is also repeated in the Q–Q plots presented in S3 Fig. We also compared our fits from the Rayleigh kernel to the more usual exponential kernel and found that the fits to China are very similar but the fit to Eswatini are slightly closer to the straight line for the Rayleigh kernel for the higher quantiles. The Akaike information criterion (AIC) values for our fits confirm the similarity between the kernels. For china that AIC for the Rayleigh kernel is 340 and exponential kernel is 343, but for Eswatini the Rayleigh kernel AIC is 1614 and exponential kernel AIC is 1607.

Our 10, 000 simulations show different realisations of the Hawkes Process model and enable us to validate our fitting. Our intuition says that these simulations represent different ways that malaria could have transmitted in alternative scenarios. Fig 5A and 5C show daily malaria case counts over time for China and Eswatini respectively. The solid red line shows observed daily cases over time and the black lines show daily cases from each simulation. There is good agreement between the simulated data and the real case counts they are fitted to, especially in the third year in China where the red line lies within the bounds of our simulations. However there are a few spikes in the first two years of China and second peak in Eswatini that we do not capture well. We are also able to separate out the cases which are

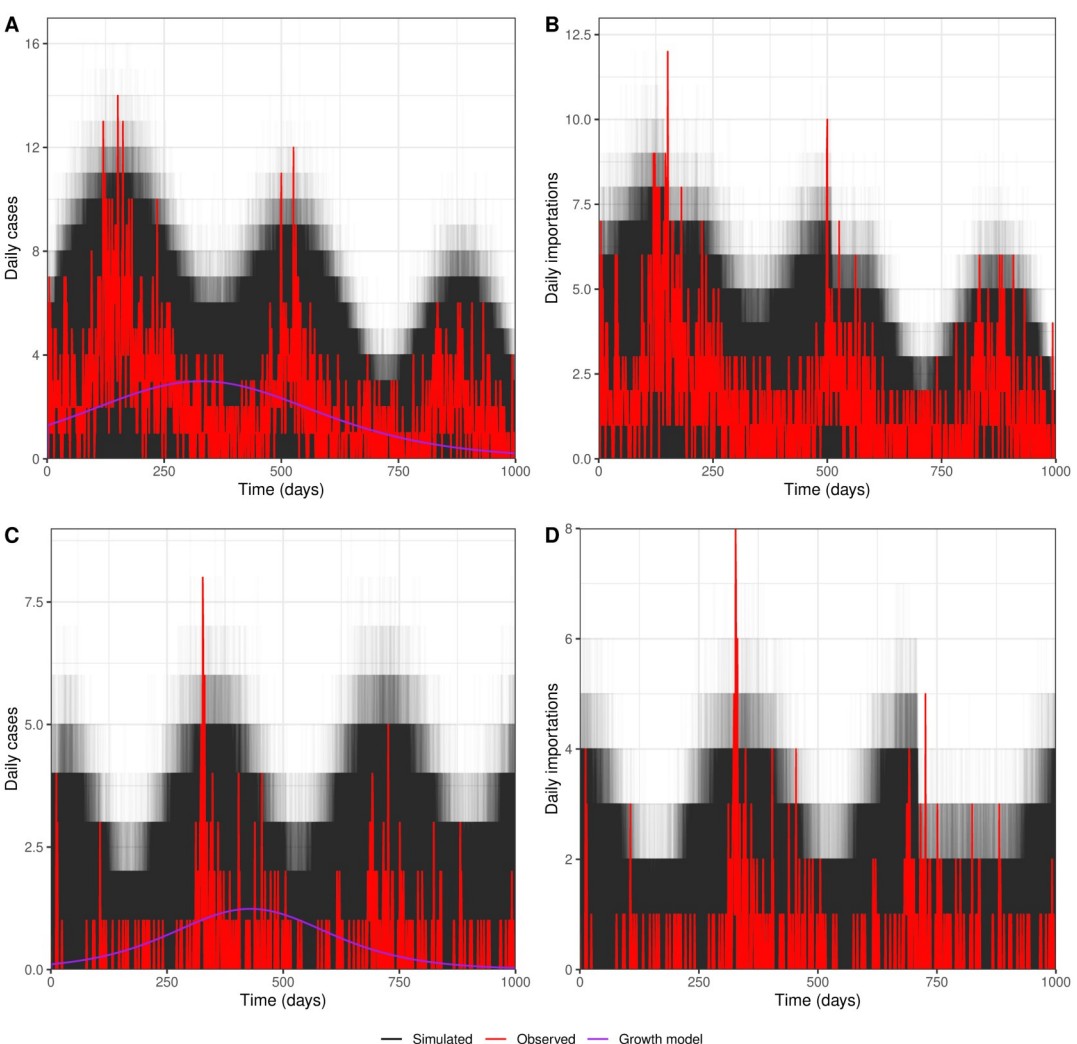

**Fig 5. Simulated daily cases for the China and Eswatini data.** Fig 5A and 5C show the daily malaria case counts for China and Eswatini respectively. The red line shows the real case counts over time and the black lines show the case counts over time from 10,000 simulations of the full fitted model. Fig 5B and 5D the daily importations for China and Eswatini respectively. Again the red line shows the real case counts over time and the black lines show simulation results.

importations from the ones that are from within country transmission, which is important in near-elimination settings. Fig 5B and 5D show the daily number of importations for China and Eswatini; again the red line shows the observed data and the black lines show our simulations. We note here that the observed importations are not necessarily determined by genetics, but usually travel history, so may not be fully accurate. We see here that the spikes we miss in the total daily cases come from importations that we do not capture well, but that we capture the seasonal trends and the general behaviour. We see this again in S4 Fig where we show the total cumulative cases and importations over time with the associated intensity. Here it is again clear that we have a good overall fit, but that we miss a few early spikes in the China data which offsets our overall importations although the year 3 behaviour is correct. We also compare our results to a simple parametric growth model and find that this model is unable to account for the seasonality in the daily malaria cases (Fig 5), although it can crudely

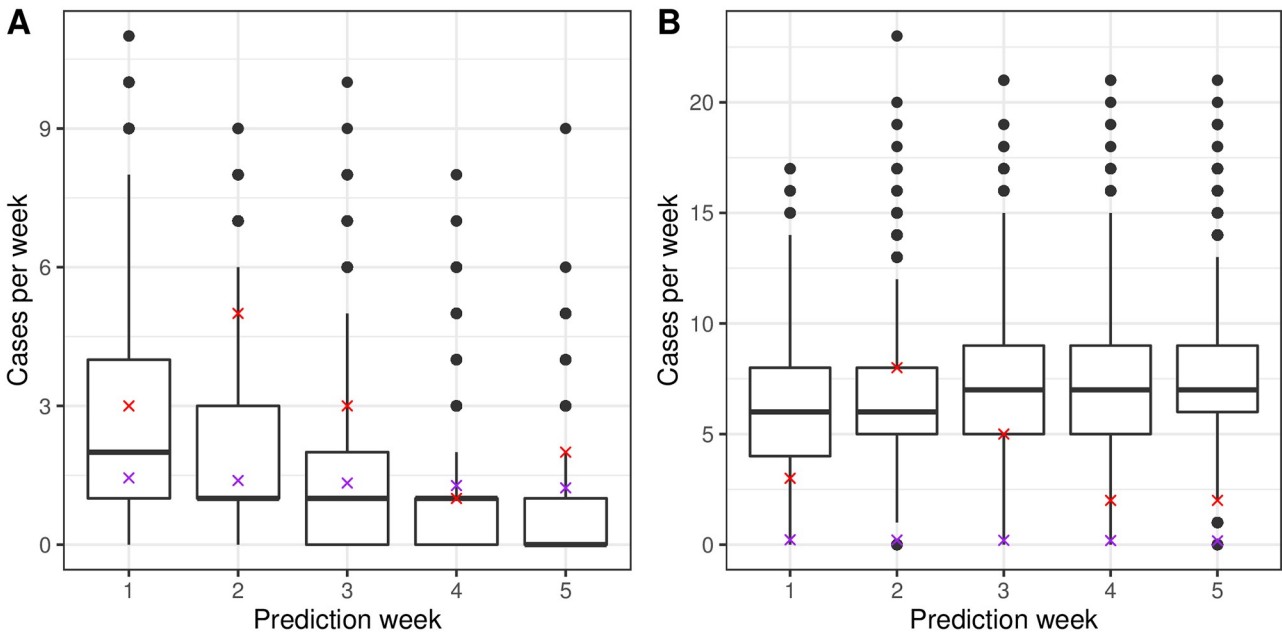

**Fig 6. Predicted total weekly cases of malaria.** Fig 6A shows weekly predicted cases of malaria for China and Fig 6B for Eswatini respectively. The red crosses show real number of cases each week, the purple crosses show the predictions from the growth model and the box and whisker plot show predictions from the 10,000 simulations. The box shows the interquartile range and the whiskers show 1.5 times the interquartile range above and below the 25th and 75th percentile.

approximate the total number of cases over the time period (S4 Fig). It also cannot be used to split out the importations from the within country transmission.

It is also possible to use Hawkes Process models to predict future cases of malaria in a country. Fig 6 shows predicted total cases in each week for the subsequent 5 weeks after we stop fitting our model. We aggregate at the weekly level because there are very few daily cases. We get good agreement between the real cases (red crosses) and the 10, 000 simulations for one month into the future for both countries, but the growth model (purple crosses) does not predict the new cases each week well in Eswatini because it predicts there is only a total of one new case during the 35 days considered (this is split over the 5 weeks since it is a continuous model). This agreement between our model and reality can also been seen in the cumulative prediction box and whisker plot in S5 Fig. However, neither the growth model fit to China or Eswatini predict well when cumulative cases are considered instead of weekly new cases. It is possible to predict further with the Hawkes Process model, but the predictions become less reliable. In particular in China, the fitted exogenous term has reached zero, meaning the simulations suggest that elimination has occurred. If we refit with more data, the $\mu(t)$ trend alters slightly and elimination is delayed.

## Discussion

Mathematical modelling is an important tool for helping countries close to eliminating malaria reach their goals. Recreating disease transmission patterns in low-endemicity settings is an important first step for validating these methods and their utility for informing policy. In this paper, we have shown that semi-mechanistic Hawkes Process models can be used to model the number of infections of malaria over time in both Yunnan Province, China, and Eswatini. We have also shown that it is necessary to make disease specific modifications to the traditional kernel to recreate malaria transmission. We estimated similar case reproductive numbers as

other methods using the same data. Routledge et al. [5] estimate a mean $R_c$ of 0.29 in 2011, 0.25 in 2012 and 0.11 in 2013, which is overlaps the confidence intervals of our estimate of 0.39 [0.23 − 0.99] for the first two years. Similarly, Reiner et al. [39] estimate the $R_c$ for Eswatini in different regions between 0.08 and 1.70, which encompasses our estimate of 0.30 [0.05 − 1.02] although our upper confidence interval is still lower than theirs. We also find that our seasonality matches the seasonality in the importations well along with the timings of the rainy seasons and travel patterns in these countries [5, 41]. These Hawkes Process methods enable us to include mechanisms of transmission that are not considered in purely statistical methods but do not need the same quality of data that is necessary for network models, as shown by the robustness of our parameter fitting to 10% missing data. Unfortunately, we do not capture the initial increase in cases towards the end of year 1 in Eswatini, caused by importations, as well as the spikes in importations in China during years 1 and 2. This could reflect policy changes, which decrease the number of importations in the subsequent years.

The use of Hawkes Processes is especially well suited to malaria modelling in near-elimination settings. This is because not only can these methods be used to recreate cases over time, which is hard to do, but they can be used to disentangle the relative contribution of importation verses local transmission where malaria control programs traditionally rely on self reported travel history that may not be accurate [42]. This is especially important in scenarios where $R_c < 1$ and malaria transmission transition from being community driven to being driven by importations. In these situations, understanding how many cases are being imported is perhaps more important to policy makers than the reproduction number, since local transmission is not sustained. This means public health bodies can target their interventions and treatment towards the demographic who travel and also potentially to the neighbouring countries where the cases are originating from. Our fits to the overall case data are better than to our importations because we choose the parameters for the Hawkes Process that minimise the error in the cumulative case counts and do not include information about travel history or which cases were imported in our fitting procedure. We choose this parameterisation for our log-likelihood because we wanted to showcase how this method could be used to ascertain the proportion of imported malaria cases when the health systems do not know how many cases originated outside the community.

A benefit of modelling malaria transmission is that we can extend our models and forecast future behaviour. We show that in both China and Eswatini our median estimated case counts matches the actual case count very well. This could provide insights to policy makers about short term transmission, which could be further improved by adding in a spatial component. From Fig 5 we see that China has very successfully managed to reduce importations over the time period studied, whereas, importations have increased slightly during the study in Eswatini.

We recognise that despite this novel implementation of the Hawkes Process method providing a flexible and useful tool for modelling malaria there are several limitations. Our method requires a unique time stamp for each individual malaria case. This is often not available in the line lists provided by the surveillance system because they are recorded by the day of presentation of symptoms. We therefore add noise to the data to recreate unique timings. We investigated the impact of adding different types of uniform or normally distributed noise to our dates but this did not impact the fits of our model significantly. We also only consider a snapshot of dates in our fit because we want to compare our forecasts of the model to true data and simulation is slow because we are solving a NP hard problem to find the maximum intensity of the Rayleigh kernel with a delay. Speeding this up is an area of ongoing research along with making this model spatial since the usual methods in e.g. Reinhart [34] did not work satisfactorily for our data set. Our optimisation surface is non-convex so care needs to be taken,

as we have, to ensure the solution returned is a true minimum and not a saddle point. Our final limitation is that we do not consider the prospect of some cases coming from previously relapsed cases instead of new infections.

## Supporting information

**S1 Text. Additional methods.**
(PDF)

**S2 Text. Directional derivatives of the negative log-likelihood.**
(PDF)

**S3 Text. Maximum intensity for a Rayleigh kernel.**
(PDF)

**S4 Text. Branching factor derivation.** Derivation of the branching factor for a Rayleigh kernel.
(PDF)

**S1 Fig. Re-fitted estimates for the how the importation intensity varies through time.** This is an un-magnified version of Fig 2B. The red line shows the importation intensity calculated from the initial parameters and the black lines shows the importation intensity calculated from the parameters fit from each simulation.
(TIF)

**S2 Fig. Impact of under-reporting on the case reproduction number.** The points show our median estimate for $R_c$ at each percentage of data fit to and the error bars show the 95% confidence intervals.
(TIF)

**S3 Fig. Comparison of goodness of fit measures for the exponential kernel (red) and Rayleigh kernel (blue) with a 15 day delay.** S3A and S3D Fig show $\Lambda(t_i)$ against i for China and Eswatini respectively, S3B and S3E Fig show Kolmogorov–Smirnov tests for China and Eswatini respectively and S3C and S3F Fig show quantile–quantile plots for China and Eswatini respectively. The solid line shows the line $y = x$ and the dashed lines show the 95% credible intervals for each test.
(TIF)

**S4 Fig. Simulated counts and intensities for the China and Eswatini data.** S4A and S4C Fig show malaria case counts for China and Eswatini respectively. The red line shows the real case counts over time and the black lines show the case counts over time from 10,000 simulations of the full fitted model. The green line shows the real case count over time from the cases labelled as importations and the blue lines show the case counts over time from 10,000 simulations of just the exogenous term (Eq (6)). S4B and S4D Fig shows the calculated Hawkes intensity (Eq (1)) for China and Eswatini respectively. The red line shows the intensity calculated from the fitted parameters and real events, whereas the black lines show the intensity calculated from the fitted parameters and the simulated events.
(TIF)

**S5 Fig. Predicted cumulative cases of malaria presented every seven days after 1000 days (the time period the model was fit to).** S5A Fig shows cumulative cases of malaria for China and S5B Fig for Eswatini respectively. The red crosses show real number of cumulative cases, the purple crosses show the predictions from the growth model and the box and whisker plot show predictions from the 10,000 simulations. The box shows the interquartile range and the

whiskers show 1.5 times the interquartile range above and below the 25th and 75th percentile.
(TIF)

**S1 Table. Values and 95% confidence intervals for model parameters fit the the China data set.** Uncertainty was calculated using the bootstrap method in Reinhart [34] and Sarma et al. [35].
(PDF)

**S2 Table. Values and 95% confidence intervals for model parameters fit to the Eswanti data set.** Uncertainty was calculated using the bootstrap method in Reinhart [34] and Sarma et al. [35].
(PDF)

## Acknowledgments

The authors would like to thank Joshua Proctor for early discussions about using Rayleigh kernels to model malaria and for his comments on the final draft. They would also like to thank Jeremy Minton for his help with the coding.

[1] https://github.com/mrc-ide/epihawkes

## Author Contributions

**Conceptualization:** H. Juliette T. Unwin, Swapnil Mishra, Samir Bhatt.

**Data curation:** Shengjie Lai.

**Formal analysis:** H. Juliette T. Unwin.

**Funding acquisition:** H. Juliette T. Unwin, Samir Bhatt.

**Investigation:** H. Juliette T. Unwin.

**Methodology:** H. Juliette T. Unwin, Isobel Routledge, Seth Flaxman, Marian-Andrei Rizoiu, Swapnil Mishra, Samir Bhatt.

**Project administration:** H. Juliette T. Unwin.

**Resources:** Samir Bhatt.

**Software:** H. Juliette T. Unwin.

**Supervision:** Samir Bhatt.

**Validation:** H. Juliette T. Unwin.

**Visualization:** H. Juliette T. Unwin.

**Writing – original draft:** H. Juliette T. Unwin.

**Writing – review & editing:** H. Juliette T. Unwin, Isobel Routledge, Seth Flaxman, Marian-Andrei Rizoiu, Shengjie Lai, Justin Cohen, Daniel J. Weiss, Swapnil Mishra, Samir Bhatt.

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
