## [Decision Letter · Decision Letter 0]

6 Oct 2020

Dear Dr Unwin,

Thank you very much for submitting your manuscript "Using Hawkes Processes to model imported and local malaria cases in near-elimination settings" for consideration at PLOS Computational Biology.

As with all papers reviewed by the journal, your manuscript was reviewed by members of the editorial board and by several independent reviewers. In light of the reviews (below this email), we would like to invite the resubmission of a significantly-revised version that takes into account the reviewers' very thorough and expert comments.

We cannot make any decision about publication until we have seen the revised manuscript and your response to the reviewers' comments. Your revised manuscript is also likely to be sent to reviewers for further evaluation.

Sincerely,

Alex Perkins

Associate Editor

PLOS Computational Biology

Tom Britton

Deputy Editor

PLOS Computational Biology

Reviewer's Responses to Questions

**Comments to the Authors:**

Reviewer #1: The review is uploaded as an attachment.

Reviewer #2: See the attached report

Reviewer #3: Review attached as PDF.

Reviewer #4: This manuscript applies temporal Hawkes process models to malaria occurrence in China and Swaziland. Such self-exciting point process models are "relatively" new for applications in infectious disease epidemiology, if "relatively" means like 10 years of research or so, and they are not yet applied frequently in this field. A recent review is given by Reinhart (2018, https://doi.org/10.1214/17-STS629), with a focus on spatio-temporal extensions of the simple Hawkes process.

My main concern with this manuscript is that the purely temporal Hawkes model presented here is somehow obsolete as spatio-temporal versions have already been established and applied, also for infectious diseases. The manuscript seems to completely *ignore* these methods and applications, concentrates on a simple temporal Hawkes model albeit saying that "malaria is a complex disease to model [...] the inoculation rate varies greatly in space". All the more important is a spatially structured model, in particular when investigating the probability of fade-out. It would be interesting to see the suggested temporal Rayleigh kernel with delay applied in a spatio-temporal model and compare it to exponential and nonparametric triggering functions, respectively.

The other main issues are potential errors in likelihood maximization and a lack of confidence intervals for the parameter estimates.

Major issues

------------

1. The manuscript suggests that Hawkes process models are "relatively". I'd argue that such models aren't that rare in the literature, certainly if we also look for more advanced spatio-temporal Hawkes models. The following research seems to have been ignored:

a) Methodological/Software-focussed: An implementation for temporal Hawkes processes is provided by the R package "PtProcess" (already ~10 years old). A multivariate temporal Hawkes process for infectious disease transmission across a network of individuals was proposed by Höhle (2009, https://doi.org/10.1002/bimj.200900050) and a spatio-temporal self-exciting process by Meyer et al (2012, https://doi.org/10.1111/j.1541-0420.2011.01684.x) with implementations in the R package "surveillance", whereas Almutiry and Deardon (2019, https://doi.org/10.1515/ijb-2017-0092) focus on individual-level and network effects and assume a time-constant triggering kernel, with implementation in the R package "EpiILMCT".

b) Applications:

https://doi.org/10.1198/jasa.2011.ap09546 (crimes, many more publications in this field)

https://doi.org/10.1080/01621459.2011.641402 (invasive plant species)

https://doi.org/10.1111/j.1541-0420.2011.01684.x (invasive meningococcal disease)

https://doi.org/10.1080/01621459.2015.1135802 (e-mail communication behaviour)

https://doi.org/10.1098/rspb.2016.0952 (spread of a wildlife pathogen)

https://doi.org/10.1080/02664763.2020.1825646 (Ebola)

From my quick search for applications, I would agree that only "a few people now use Hawkes Processes *for epidemiological modelling*", but the modelling approach per se is really no longer in its infancy. Furthermore, there are examples of using a seasonal exogeneous effect (just like this) in the literature (p. 5, l. 160), e.g., in the aforementioned meningococcal disease application.

2. The manuscript mentions Kelly et al (2019) for a recent Hawkes process modelling approach. A nice feature of that work is that no particular functional form is assumed for the triggering kernel; instead a step function is estimated and smoothed. The authors should really consider such a nonparametric approach as a means of validating the Rayleigh kernel parametrization.

3. The authors suggest to use a kernel with delay to account for the latent period. The current approach has two problems:

a) the kernel is exactly 0 until day 12, when it experiences a sharp increase. A smooth increase seems to be more realistic, in particular because these 12 days won't hold for every case.

b) for the kernel to make sense, wouldn't the dates t_i need to correspond to the day the person got infected? The authors say in the discussion that "they are recorded by the day of presentation of symptoms".

4. I was really surprised to read that numerical log-likelihood maximization suffered from convergence problems for this model (the parameter space isn't really "complex", l. 332) and these relatively large data sets. I suspected that the gradient might be wrongly derived or implemented. As it turns out, the analytic gradient implemented for `delta` disagrees with a numerical approximation. Running `vignette("fitting")` from the authors' R package and then

set.seed(1)

par <- c(alpha = runif(1, 0, 1), delta = runif(1, 0, 1), A = runif(1, 0, 1), B = runif(1, 0, 1))

maxLik::compareDerivatives(neg_log_likelihood, ray_derivatives, t0 = par, events = events, delay = delay, kernel = ray_kernel, mu_fn = mu_fn, mu_diff_fn = mu_diff_fn,mu_int_fn = mu_int_fn)

I get

t0

alpha delta A B

0.2655087 0.3721239 0.5728534 0.9082078

analytic gradient

[,1] [,2] [,3] [,4]

[1,] 217.0904 217.0904 26.45247 566.8895

numeric gradient

alpha delta A B

[1,] 217.0904 -143.9715 26.45247 566.8895

Note that I could only investigate this further because the code was submitted (published, actually) together with the manuscript!

What a nice example for the advantage of open science with open source software. :)

I'm curious if the convergence problems go away when the gradient is validated.

5. Related to the above: The authors state that the likelihood loss function is non-convex, referencing Kong et al (2019). I couldn't find this information in the referenced paper. Please verify. From what I know from Rathbun (1996), the log-likelihood of a self-exciting point process is concave if the CIF is linearly parametrized.

6. Again related: Please re-check the model fit and simulation based on the exponential kernel. The finding that simulations from the exponential kernel didn't recreate the data may well suffer from a similar error.

7. The authors seem to have been careful not to write about *basic* reproduction numbers but "case reproduction numbers". I think it is really worth noting that reproduction numbers estimated from such a branching process with immigration are "adjusted" for infections occurring independently of previously *observed* infections. Please see the discussion of Delamater et al (2019, https://doi.org/10.3201/eid2501.171901) on the importance of communicating what is meant by R. Furthermore, from the referenced work by Routledge et al. it seems that the case reproduction number is decreasing over the years. Have you considered estimating case-specific effects on the triggering rate (as in seismology and in some of the aforementioned point process approaches in the literature) as to model decreasing magnitudes $\\alpha$ (and thus R) over the years?

8. Given that inference on the reproduction number is of scientific interest, its estimate should really be accompanied by a 95% confidence interval to quantify uncertainty. Different methods have been proposed to estimate the variance-covariance matrix of the MLE in such point process models (see, e.g., the aforementioned review by Reinhart). I think for your model a numerical estimate of the Hessian would provide a reasonable basis for confidence intervals (after the analytical gradient has been corrected and validated).

9. Isn't underreporting also an issue for Malaria cases? The endogenous contribution will be underestimated if cases are missing in the line list. Statistical inference requires knowledge about the data-generating process; a sophisticated Hawkes model can be useful, but underreporting can bias the parameter estimates.

Minor issues

------------

- The introductory part (including the background section) is relatively long. It reminds me of a textbook or thesis chapter. I'm sure some parts could be shortened. For example, Ogata's thinning algorithm is well known and doesn't need to be explained in detail, at least not as part of the main text. The crucial part is that it requires a (temporary) upperbound for the conditional intensity.

- The introduction says that mechanistic models "may make strong assumptions such as the homogeneity of the population". I don't see that this is avoided by the proposed Hawkes model, especially because it is spatially aggregated.

- p. 2, l. 35-36: Kelly et al. had a constant $\\mu$ backgroundin their model so it is wrong to says they used a "model with just an endogenous term".

- Eq. 2: $\\lambda^H$ -> $\\lambda^H(t)$

- SI part 1 wouldn't be necessary (you could also just reference a textbook) but is nice to have.

- SI Eq. 31: $\\partial M$ -> $\\partial N$.

- Eq. 5: $\\tau$ -> $d \\tau$

- p. 5, l. 158: Is it true that the upper bound is "no longer trivial to find" just because a constant delay is introduced? Or is non-monotonicity a problem (as suggested in line 198)? It seems to me that the mode of the Rayleigh kernel could still be used, i.e. assuming the value $\\phi(1/\\sqrt{\\delta})$ for all currently infectious individuals ($t_i < t + \\Delta$). I understand that the time-varying exogeneous term complicates the choice of a suitable upper bound.

- p. 6, l. 201: delays are denoted by $\\tau$ here but by $\\Delta$ in eq. 8

- p. 6, l. 171: please explain Plasmodium vivax.

- The data is first mentioned on page 6, but the reader has to wait for Figure 3 on p. 9 to finally see what data we were actually modelling. I always prefer to see the data or a descriptive summary thereof, before thinking about any modelling strategy. I'd suggest to describe the data together with the goal of the analysis earlier in the manuscript.

- Maybe I've overlooked it, but please mention the total number of infections of the two datasets in the text. I only found out approx. from Figure 3.

- p. 8, l. 227: \\alpha = \\delta = 0 contradicts the parameter definition in Eq. 7.

- Figure 2: The term "serial interval distribution" is misleading in that we don't see a density; the kernel doesn't integrate to 1. (This one is picky, I know. Just ignore this point if you prefer.)

Sebastian Meyer

**Have all data underlying the figures and results presented in the manuscript been provided?**

Reviewer #1: None

Reviewer #2: Yes

Reviewer #3: **No: **The authors indicated data is available via their GitHub, but no data appears to be present there, only code. Code reproducing their figures is also not available. I was not able to find the line-level data by chasing up references, either. It should be included here or deposited in a publicly available source. If it's already publicly available, that should be made clearer in the text.

Reviewer #4: **No: **The data is not available from the referenced GitHub repository. Is it possible to publish the timings + importation status for the two data sets (including the added noise)? I guess this would be sufficient for reproducibility.

PLOS authors have the option to publish the peer review history of their article (what does this mean?). If published, this will include your full peer review and any attached files.

Reviewer #1: No

Reviewer #2: No

Reviewer #3: No

Reviewer #4: No
---

## [Decision Letter · Decision Letter 1]

20 Jan 2021

Dear Dr Unwin,

Thank you very much for submitting your manuscript "Using Hawkes Processes to model imported and local malaria cases in near-elimination settings" for consideration at PLOS Computational Biology. As with all papers reviewed by the journal, your manuscript was reviewed by members of the editorial board and by several independent reviewers. The reviewers appreciated the attention to an important topic. Based on the reviews, we are likely to accept this manuscript for publication, providing that you modify the manuscript according to the review recommendations.

Sincerely,

Alex Perkins

Associate Editor

PLOS Computational Biology

Tom Britton

Deputy Editor

PLOS Computational Biology

[LINK]

Reviewer's Responses to Questions

**Comments to the Authors:**

Reviewer #1: Comments uploaded as an attachment.

Reviewer #2: I am pleased with your answers to my comments.

Reviewer #3: Attached as pcompbiol-revision-1.pdf.

Reviewer #4: The manuscript has greatly improved thanks to the many reviewers' thoughtful comments. The simulation study with its assessment of underreporting is very useful.

I'm happy with most replies to my comments and only have some minor follow-up remarks:

1. I agree that a purely temporal Hawkes model is a suitable starting point for the development of more complex spatio-temporal formulations such as "twinstim" of [26]. Purely temporal models are much faster to estimate as they don't require heavy cubature over space to evaluate the log-likelihood. FWIW, it is relatively straightforward to supply different parametric kernels in "twinstim" such as the Rayleigh kernel. I'm happy to help if you would like to use "twinstim" for comparison in the future. However, in my experience, reliable estimation of the temporal kernel in a spatio-temporal model requires a lot of events because the spatial decay reduces the effective number of events contributing to the likelihood. The Eswatini data seem to be too sparse for that, in particular if event locations are partially unknown.

2. The authors say they now reference Menon and Lee (2018) and Lime and Choi (2018). However, I couldn't find these references and the comment on the negative log-likelihood being potentially non-convex seems gone as well?

3. IMO, Fig S1 would rather suggest that the exponential and Rayleigh kernels fit equally well. I cannot see a relevant difference. Why not report the AIC values to compare the two fits?

Sebastian Meyer

**Have all data underlying the figures and results presented in the manuscript been provided?**

Reviewer #1: None

Reviewer #2: Yes

Reviewer #3: Yes

Reviewer #4: **No: **The data is not available from the referenced GitHub repository, only code and fake data.

PLOS authors have the option to publish the peer review history of their article (what does this mean?). If published, this will include your full peer review and any attached files.

Reviewer #1: No

Reviewer #2: No

Reviewer #3: No

Reviewer #4: No
---

## [Editor Report · Decision Letter 2]

23 Feb 2021

Dear Dr Unwin,

We are pleased to inform you that your manuscript 'Using Hawkes Processes to model imported and local malaria cases in near-elimination settings' has been provisionally accepted for publication in PLOS Computational Biology.

Best regards,

Alex Perkins

Associate Editor

PLOS Computational Biology

Tom Britton

Deputy Editor

PLOS Computational Biology

---

## [Editor Report · Acceptance letter]

29 Mar 2021

PCOMPBIOL-D-20-01373R2 

Using Hawkes Processes to model imported and local malaria cases in near-elimination settings

Dear Dr Unwin,

I am pleased to inform you that your manuscript has been formally accepted for publication in PLOS Computational Biology. Your manuscript is now with our production department and you will be notified of the publication date in due course.

With kind regards,

Katalin Szabo
